# Unraveling the Roles of Macrophages in Vascularized Composite Allotransplantation

**DOI:** 10.3390/biomedicines13061425

**Published:** 2025-06-10

**Authors:** Hui-Yun Cheng, Madonna Rica Anggelia, Cheng-Hung Lin

**Affiliations:** 1Center for Vascularized Composite Allotransplantation, Chang Gung Memorial Hospital at Linkou, Kweishan, Taoyuan 333, Taiwan; 2Department of Plastic and Reconstructive Surgery, Chang Gung Memorial Hospital at Linkou, Kweishan, Taoyuan 333, Taiwan

**Keywords:** macrophage, vascularized composite allotransplantation, M1 macrophage, M2 macrophage, regulatory macrophage, ischemia–reperfusion injury, transplantation rejection, transplantation tolerance

## Abstract

The phenotypic heterogeneity and functional diversity of macrophages have been increasingly appreciated, particularly regarding their roles as innate immune cells in shaping transplantation outcomes. However, their functions in vascularized composite allotransplantation (VCA) remain underexplored. In this review, we first describe the development of macrophages and the heterogeneity of macrophage differentiation, then present current insights into macrophages’ involvement across key stages of VCA, including ischemia–reperfusion injury at the peri-transplantation stage, and the outcomes following transplantation, including acute rejection, chronic rejection, and development of transplantation tolerance. The existing evidence supports that macrophages significantly influence both short- and long-term VCA graft survival. The presence of vascularized bone marrow within some VCA grafts further suggests the involvement of donor bone marrow-derived macrophage population and adds another layer of complexity to immune dynamics. Collectively, current understanding highlights the macrophage as a promising target for therapeutic intervention and warrants continued investigation into their diverse functions and potential for improving VCA outcomes.

## 1. Introduction

Macrophages are key components of the innate immune system, well recognized for their ability to detect and eliminate pathogens [1]. They serve as the frontline defenses against foreign stimuli through phagocytosis and the release of inflammatory mediators. Additionally, macrophages contribute to adaptive immune responses by processing and presenting antigens and providing costimulatory signals [2]. In transplantation immunology, macrophages can constitute up to 60% of the cellular infiltrate in rejecting allografts [3], underscoring their prominent role in alloimmune responses. However, their functional involvement in alloimmunity remains complex and not fully elucidated, largely due to their remarkable plasticity and heterogeneity [4].

Vascularized composite allotransplantation (VCA), which involves the transplantation of functional units such as hands or faces, has emerged as an important reconstructive approach to treat defects that cannot be mended by autologous tissues. Typical VCA grafts are composed of multiple tissue types, such as skin, muscles, nerves, tendons, and bones, and may elicit unique immunological responses. Notably, approximately 85% of clinical VCA recipients experience acute rejection (AR) within the first year post-transplantation [5]. AR typically manifests as erythema, edema, and maculopapular rash localized to the graft. It is primarily mediated by cytotoxic T cells and characterized by perivascular lymphocytic infiltration, epidermal dyskeratosis, and occasionally endothelialitis [6]. Some patients even endure multiple rejection episodes despite consistent immunosuppressive therapy [7]. Current rejection management strategies include adjustments to systemic immunosuppressants, combined with topical treatments, steroid bolus, thymoglobulin, or rituximab [7,8]. However, repeated rejection episodes and treatment cycles have been linked to progressive deterioration of VCA function and appearance over time [9].

Some patients develop chronic rejection (CR), which progresses slowly and is typically irreversible. Unlike AR, CR often lacks overt inflammatory signs. The pathological hallmarks of CR include graft vasculopathy that is characterized by accelerated arteriosclerosis with smooth muscle cell proliferation and inflammatory infiltration, leading to vascular narrowing and compromised blood flow. Intimal hyperplasia, the thickening of arterial walls due to smooth muscle cell migration and proliferation, occurs alongside progressive tissue fibrosis involving excessive collagen deposition in dermal and muscular compartments. These processes result in the loss of adnexal structures such as hair follicles and nails, ultimately causing ischemic damage and tissue necrosis. CR remains challenging to diagnose and currently has no effective treatment options [10].

Furthermore, animal studies reveal that CR-like vascular changes, such as intimal hyperplasia and muscular functional decline, can arise following repeated immunosuppressant bolus treatments [11,12]. These clinical and experimental findings underscore the need to optimize rejection prevention and management approaches or to develop novel therapeutic modalities.

Recent advances in transplantation immunology have continued to uncover the complex roles of innate immune leukocytes, such as macrophages, in shaping allograft outcomes. In addition to their classical function in pathogen defense, subtypes of macrophages have been implicated in processes such as graft fibrosis and immune tolerance. Their remarkable plasticity and functional diversity suggest that macrophages may play critical roles in both VCA and solid organ transplantation [4,13]. Accordingly, this review focuses on the involvement of macrophages in VCA, summarizing the current understanding of their roles across various stages and exploring potential future directions in VCA.

## 2. Overview of Leukocyte Differentiation and Function

All leukocytes, or white blood cells, are central components of the immune system that coordinate defenses against pathogens and foreign antigens. They are derived from hematopoietic stem cells (HSCs) residing primarily in the bone marrow. Early in development, HSCs differentiate into two principal lineages: lymphoid and myeloid.

The lymphoid lineage includes T cells, B cells, and natural killer (NK) cells, and plays central roles in adaptive immunity. T cells mediate cellular immunity through cytotoxic and helper functions, regulating immune responses and eliminating infected or abnormal cells. B cells are responsible for humoral immunity by producing antigen-specific antibodies that neutralize pathogens and facilitate their clearance. NK cells are another population of cytotoxic lymphocytes that play essential roles in the early defense against virally infected and transformed cells by inducing apoptosis through perforin and granzyme release. They can also modulate innate and adaptive immune responses through the production of cytokines such as IFN-γ [14].

The myeloid lineage includes granulocytes (neutrophils, basophils, and eosinophils), monocytes, macrophages, and dendritic cells (DCs), all of which play essential roles in innate immunity, inflammation, and tissue repair [15]. Granulocytes are characterized by the presence of cytoplasmic granules containing enzymes and antimicrobial substances. Among them, neutrophils rapidly respond to infection through phagocytosis and the release of reactive oxygen species (ROS) and neutrophil extracellular traps (NETs) [16]; eosinophils and basophils respond to parasitic infections and allergic reactions.

DCs are professional antigen-presenting cells. They capture antigens from peripheral tissues and migrate to lymph nodes to prime naïve T cells and initiate adaptive immune responses. DCs can be differentiated from a specific progenitor population (common dendritic progenitor) or monocytes.

Monocytes are a crucial population among the terminally differentiated myeloid cells, with features of large size, kidney-shaped nuclei, and specific cell surface markers. They circulate in the bloodstream and can migrate into tissues. Depending on the cytokine milieu and tissue context, monocytes further differentiate into DCs or macrophages. Macrophages are highly adaptable immune cells that are distributed throughout nearly all tissues. The differentiation of monocytes into macrophages is a complex process regulated by various factors that orchestrate gene expression profiles for macrophages to perform specialized roles. For example, monocytes respond to macrophage colony-stimulating factor (M-CSF) and adopt macrophage phenotypes characterized by enhanced phagocytic capacity and anti-inflammatory cytokine production, promoting resolution of inflammation. In contrast, granulocyte-macrophage colony-stimulating factor (GM-CSF) promotes the differentiation of macrophages specialized in antigen presentation and activation of adaptive immunity [17]. Specific transcription factors such as PU.1 and IRF8 are critical factors to regulate these responses [18,19]. Macrophages display remarkable phenotypic and functional heterogeneity, which will be described in detail in the next section.

## 3. Heterogeneity of Macrophages

Macrophages can be differentiated either from circulating monocytes originating in the bone marrow or from specific embryonic precursors. They exhibit remarkable heterogeneity, dynamically adapting to environmental cues [1]. Traditionally, macrophages are considered to polarize into two primary activation states: the classically activated M1 phenotype and the alternatively activated M2 phenotype. In addition to M1 and M2 macrophages, two other subtypes of macrophages—regulatory macrophages (Mregs) and tissue-resident macrophages (TRMs)—have demonstrated key roles in alloimmunity (Figure 1).

### 3.1. M1 Macrophage

M1 macrophages play critical roles in pathogen defense and the initiation of inflammatory processes. They can be differentiated from naïve macrophages (M0) through engagement of pattern recognition receptors (PRRs) by damage-associated molecular patterns (DAMPs) or pro-inflammatory signals such as interferon-γ (IFN-γ) and lipopolysaccharide (LPS) [20]. M1 macrophages are characterized by the production of pro-inflammatory cytokines, including tumor necrosis factor-α (TNF-α) and interleukin-1β (IL-1β), and generation of ROS and nitric oxide [21], which contribute to cytotoxicity and tissue damage [22,23]. These macrophages typically express major histocompatibility complex (MHC) class II and costimulatory molecules such as CD80 and CD86, which facilitate antigen presentation and promote T helper cell 1 (Th1) differentiation [24]. The IFN-γ secreted by Th1 cells further reinforces M1 polarization. In transplantation contexts, M1 macrophages have been associated with acute rejection, as their pro-inflammatory mediators can amplify alloimmune responses and contribute to graft injury [25].

### 3.2. M2 Macrophage

M2 macrophages were first identified as being driven by anti-inflammatory cytokines such as IL-4 and IL-13, with the capacity to secrete anti-inflammatory IL-10 and transforming growth factor-β (TGF-β), which promote resolution of inflammation, tissue repair, wound healing, and tissue regeneration [26]. However, excessive activation of M2 macrophages has also been implicated in tissue fibrosis and chronic graft dysfunction [27,28]. M2 macrophages are characterized by the markers including CD206, CD163, arginase 1, and IL-10 (14).

The M2 macrophages have been categorized further into four subtypes: M2a, M2b, M2c, and M2d, based on distinct functional marker expression. M2a macrophages, stimulated by Th2 cytokines IL-4 and IL-13, are involved in tissue repair and exhibit enhanced endocytic activities [29]. M2b macrophages, activated by immune complexes and Toll-like receptor (TLR) ligands, contribute to Th2 immune responses and help regulate fibrosis [30,31]. M2c macrophages, induced by IL-10, are associated with immunoregulation and tissue remodeling [30]. M2d macrophages, also known as tumor-associated macrophages, exhibit strong angiogenic potential and suppress anti-tumor T cell response, promoting tumor progression [32,33]. However, the conversion between M2 macrophage subtypes under specific stimuli has been reported, demonstrating the plasticity of these cells [34,35].

Recent studies suggest that macrophage activation is not a binary process but rather exists along a spectrum of functional states, where the M1 and M2 macrophages represent the two ends, and can be influenced by cytokine milieu, metabolic pathways, and epigenetic modifications [36,37]. The single-cell RNA sequencing (scRNAseq) study has revealed hybrid phenotypes that express markers of both M1 and M2 macrophages, indicating a continuum of activation states rather than distinct subtypes [38]. Moreover, macrophage polarization is dynamic and can shift in response to environmental cytokines and metabolic stimuli. For example, Chen et al. demonstrated that enhanced glycogen metabolism in M2 macrophages transformed them to an M1-like phenotype, ultimately alleviating cardiac fibrosis [39]. Conversely, exosomes derived from M2 macrophages induced M1-to-M2 conversion and promoted cutaneous wound healing [40]. Dysregulation of M1/M2 balance has been implicated in various disease processes, including atherosclerosis and cancer, where an imbalance may exacerbate pathology [41,42,43].

### 3.3. Regulatory Macrophage

Regulatory macrophages (Mregs) represent another distinct population differentiated from bone marrow-derived monocytes. They express costimulatory molecules and possess strong antigen presentation capacity while producing high levels of anti-inflammatory IL-10 and low levels of pro-inflammatory IL-12 [44]. Mregs can be induced by M-CSF and IFN-γ [45], or converted from M1 or M2 macrophages under specific conditions [46].

Mregs exert potent immunoregulatory effects by suppressing allogenic T cell responses via indoleamine 2,3-dioxygenase (IDO) or nitric oxide production, as well as by inhibiting IL-2 secretion and promoting phagocytosis [47]. Coculturing Mregs with allogenic T cells promotes differentiation and expansion of regulatory T cells (Tregs), potentially mediated by TGF-β secreted by Mregs [48].

Mregs have been explored as a cellular therapy in transplantation. Infusion of donor-derived Mregs prior to transplantation significantly prolonged allograft survival in murine cardiac transplantation models, and these effects were further potentiated when combined with a short course of rapamycin (10 days) [49]. In clinical studies, autologous Mregs have been associated with the long-term survival of allogeneic skin grafts [50]. Hutchinson et al. demonstrated that Mreg therapy in renal transplant patients is feasible, safe, and allows tapering of conventional immunosuppression to tacrolimus monotherapy [51,52].

### 3.4. Tissue-Resident Macrophage (TRM)

TRMs originate from embryonic progenitors and are specialized cells that permanently inhabit specific tissues. Unlike the monocyte-derived macrophages, which migrate into tissues in response to inflammatory stimuli and subsequently polarize into M1, M2, or Mreg phenotypes, TRMs are capable of self-renewal and longevity. They are finely tuned by local environmental cues, maintaining local tissue homeostasis and immune surveillance.

TRMs are widely distributed across tissues, including microglia in the central nervous system, Kupffer cells in the liver, and Langerhans cells in the skin. TRMs are also found in the spleen, gut, lung, lymph nodes, and adipose tissues [4]. Research has revealed that TRMs are guided by a shared core transcription factor program while acquiring tissue-specific adaptations that define their phenotype and function [53]. Many TRMs possess M2-like attributes, contributing to tissue repair and immune tolerance. Some TRMs were shown to facilitate angiogenesis and suppress fibrosis [54]. On the other hand, monocyte-derived macrophages may convert to TRM-like phenotypes with specific stimuli [55,56]. Emerging evidence supports the role of TRMs in the induction of transplantation tolerance, potentially through M2-like polarization and the clearance of apoptotic T cells [57]. For instance, in murine heart transplants, depletion of TRMs accelerated graft rejection, underscoring their protective role in transplantation immunology [58].

### 3.5. Emerging Insights into Macrophage Heterogeneity During Transplantation Rejection

Recent strides in technical developments, such as high-dimensional tissue profiling [59], spatial proteomics and transcriptomics [60,61], mass cytometry [62], together with the previously discussed scRNAseq, now permit precise cellular characterization of rejecting transplants at single-cell resolution. These technologies have uncovered previously unrecognized cell populations [63,64] and revealed complex cellular composition in rejecting grafts, including diverse macrophage subsets [65]. For example, Barbetta et al. analyzed liver biopsies diagnosed as no rejection (NR), T cell-mediated rejection (TCMR), and chronic rejection (CR). They identified four M1 and five M2 macrophage subsets. While overall M1 (CD68^+^CD163^low^) macrophages were elevated in TCMR and M2 (CD68^+^CD163^high^) remained relatively unchanged across groups, specific M2 subsets showed differential behaviors: CD16^+^M2 decreased while HLA-DR^+^M2 increased with significance in the TCMR compared to the NR group, highlighting functional heterogeneity within the M2 category [60]. In a rejecting mouse heart allograft, five macrophage subtypes were described among the total of 21 cell types, including two TRM populations and two infiltrating M1-like macrophages. One infiltrating macrophage expressed high levels of costimulatory *CD40* and other genes associated with rejection, while another showed a quiescent metabolic profile and may represent a primed but inactive state [66]. Similarly, six macrophage/monocyte cell clusters were identified in rejecting kidney transplants, with the *Ly6C^low^Ear2^+^* and *Ly6C^low^Mrc1^+^* macrophages implicated in the transition from acute rejection to chronic rejection [67]. Furthermore, scRNAseq analysis revealed changes in TRMs in the kidney from a healthy state to rejection, including suppressed transendothelial migration and reduced phagocytosis [68].

Together, these findings emphasize that even within the defined macrophage classifications like M1 and M2, nuanced subpopulations exist with distinct functional roles. In light of accumulating evidence, the growing interest in macrophage heterogeneity is a natural progression, reflecting an emerging recognition of their critical influence on transplant outcomes.

## 4. Current Understanding of Macrophages in VCA

### 4.1. Overview of VCA Immunobiology

VCA initiates a complex sequence of cellular and immunological events that begin immediately upon graft reperfusion. Here we provide an overview of these events, and more details can be found in the literature dedicated to this topic [6,69]. Ischemia–reperfusion injury (IRI) is the first challenge encountered and is an inevitable consequence of transplantation surgery. During graft procurement and vascular anastomosis, temporary cessation of blood flow results in oxygen deprivation, energy depletion, increased lactate production, and intracellular acidification. These metabolic disturbances ultimately lead to osmotic imbalance, cell swelling, apoptosis, and tissue damage [70,71]. Upon reperfusion, oxidative stress further exacerbates injury by disrupting mitochondrial integrity and triggering the release of DAMPs. These molecules activate PRRs on innate immune cells, including macrophages, DCs, and neutrophils, leading to the production and release of proinflammatory cytokines such as TNF-α, IL-1β, and IL-6. Consequently, a proinflammatory microenvironment is created, accompanied by enhanced alloreactivity through upregulated expression of MHC class II molecules [72]. Neutrophils are recruited to the graft site, where they may exacerbate tissue damage through degranulation and ROS release [73,74].

Monocytes are subsequently recruited and differentiate into macrophages within the graft. These macrophages may adopt either proinflammatory M1 or anti-inflammatory/tissue-reparative M2 phenotypes depending on the local cytokine milieu. The M1 polarization reinforces inflammation through further cytokine secretion and antigen presentation. If inflammation persists, DCs and activated macrophages migrate to draining lymph nodes, priming alloantigen-specific T cells and initiating adaptive immunity. Cytotoxic CD8^+^ T cells and CD4^+^ helper T cells infiltrate the graft and mediate AR through direct cytolysis and cytokine-mediated injury [75].

CD4^+^ T cells can differentiate into subsets such as Th1 cells that produce IFN-γ to further amplify macrophage activation, and Th17 cells that promote neutrophil recruitment and tissue inflammation. Tregs, conversely, may suppress effector T cell responses and promote transplantation tolerance, especially under immunomodulatory conditions. CD8^+^ T cells recognize donor MHC class I molecules and induce apoptosis of donor cells via perforin-granzyme pathways and Fas-FasL interactions [75].

B cells contribute to VCA rejection by producing donor-specific antibodies (DSAs), which activate the classical complement cascade and promote antibody-dependent cellular cytotoxicity (ADCC), primarily targeting graft vasculature [76]. B cells additionally function as antigen-presenting cells and cytokine producers, modulating T cell activation and differentiation. At this stage, NK cells may also participate in mediating cytotoxicity against graft cells lacking self-MHC and contributing to both early injury and chronic vasculopathy [77,78].

Over time, repeated injury or inadequate resolution of inflammation can lead to CR as described previously. Alternatively, efforts to induce donor-specific tolerance have shown promising success in preclinical models, for example, through costimulatory blockade [79] and the infusion of mesenchymal stromal cells (MSCs) or Tregs [80,81,82]. Additionally, the vascularized bone marrow in certain types of VCA has demonstrated potent tolerogenic effects, further enhancing the potential of establishing transplantation tolerance [83,84].

Although macrophages have not been a major focus in VCA research, the above overview has implicated that macrophages may play roles in different stages of VCA. Evidence has demonstrated their participation in IRI, acute rejection, and chronic rejection of animal models and clinical settings. Furthermore, one type of Mreg, the transplant acceptance-inducing cell (TAIC), has been applied to a rat hindlimb VCA model, where its efficacy in prolonging VCA graft survival has been evaluated [85]. Recent progress on macrophages in VCA is briefly discussed below.

### 4.2. Macrophages in Ischemia-Reperfusion Injury

VCA grafts are sensitive to IRI since they contain highly metabolically active tissues such as muscles [86]. Friedman et al. investigated surgery-related injuries, including IRI, by comparing the responses of allogeneic to syngeneic grafts. Their study demonstrated that macrophage infiltration occurred as early as postoperative day (POD) 2, initially at the donor–recipient interface and progressively expanding linearly to the outer perimeter by POD 5 in allogeneic grafts. In contrast, in syngeneic grafts, macrophage infiltration remained confined within the graft by POD 5. These findings suggest that surgical damage can initiate an inflammatory cascade that may facilitate adaptive immune responses and rejection in an allogeneic environment [87].

In reality, grafts often require transport before transplantation surgery, necessitating preservation techniques such as static cold storage in preservation solution or machine perfusion systems [86,88]. Nevertheless, various degrees of IRI still occur during this process. For instance, Datta et al. examined the impact of short (1-hr) versus long (6-hr) cold ischemia on murine hindlimb osteomyocutaneous grafts prior to transplantation. Both groups exhibited elevated macrophage infiltration and increased proinflammatory cytokines in serum. However, recipients in the long ischemia group experienced significantly higher cytokine levels in graft muscle and kidney injury, leading to a higher fatality rate, suggesting that long ischemia exacerbated inflammatory responses, potentially triggering a cytokine storm and systemic damage [73]. Similarly, in a rat VCA model, 7 h cold ischemia resulted in severe vessel fibrosis and intimal hyperplasia by POD 60, accompanied by increased infiltration of macrophages and T cells compared to grafts transplanted without cold ischemia [89]. These pathological changes resembled CR, underscoring the prolonged impact of IRI and the crucial role of macrophages in this process.

### 4.3. Macrophages in Acute Rejection of VCA

Macrophages, particularly the M1 subtype, have been implicated in acute transplantation rejection [90,91], though the M2 subtype has also been shown to be involved in antibody-mediated rejection, highlighting the multifaceted roles of macrophages. However, their specific contributions to the acute rejection of VCA remain less characterized. Increased macrophage infiltration has been observed in both facial and hand VCA grafts during clinical rejection, although detailed phenotypic analyses have not been performed. Hautz et al. reported that CD68^+^ macrophages constituted approximately 10% of the perivascular infiltrate and were scattered throughout the interstitial dermis in grade I rejection of hand allografts [92]. The level of CD68 was correlated with rejection severity from grade I to IV, but the correlation coefficient was low (0.226). CD68^+^ macrophages were also detected in biopsies that did not show rejection symptoms [93]. Baran et al. reported similar findings, demonstrating higher numbers of CD68^+^ macrophages, along with CD4^+^ and CD8^+^ cells, in the hand graft skin compared to the recipient’s own skin, even under the stable condition without rejection. Langerhans cells, a subset of TRMs, were significantly more abundant in the graft skin [94], suggesting that the allograft environment may promote their proliferation locally. Understanding the role of these cells in maintaining the graft stability warrants further investigation.

Functional insights into macrophages in VCA rejection have emerged from animal studies. Zhang et al. performed bulk RNAseq on rejecting swine VCA grafts and demonstrated significant upregulation of PRRs and DAMP-related genes, highlighting the importance of innate immune activation in rejection. Using a gene deconvolution bioinformatics tool [95], the M1 and M2 macrophages were characterized as the top-ranked immune cell types, alongside *CD8^+^* T cells, in rejecting grafts [96].

C-reactive protein (CRP) is an acute-phase protein generated in response to inflammation and tissue damage. It binds to injured cells, enhances leukocyte recruitment, and activates the complement cascade [97]. When the clinical facial transplantation patients experienced rejection, their serum CRP levels were significantly increased, linking CRP to VCA rejection [98]. Kiefer et al. administered active form of human CRP (pCRP) to rats that were transplanted with allogenic (VCA) or autologous (replantation) grafts and discovered that the VCA recipients showed dense CD68^+^ infiltration derived from the recipient pro-inflammatory CD43^low^HIS48^high^ monocytes on POD 3 to 5, whereas the autologous group showed less degree of infiltration and peaked on POD 3. In contrast, both groups did not show apparent CD4 or CD8 lymphocyte infiltration on POD3. Depleting the monocytes and macrophages with clodronate liposome significantly prolonged the VCA survival, confirming their pivotal role in IRI and the alloantigen-induced immune response [98].

Focusing on the injury at the early stage following transplantation, Sommerfeld et al. applied a commercially available porcine urinary bladder-derived extracellular matrix (ECM) biomaterial (MatriStem, Acell) around the approximation site between donor and recipient musculature after surgery, and found that it enhanced the efficacy of CTLA4-Ig costimulatory blockade and rapamycin in prolonging VCA graft survival. MatriStem has been clinically shown to promote tissue repair, such as in conditions of chronic wounds [99] and flap salvage [100]. When applied to the murine VCA, the presence of MatriStem ECM induced a conversion of proregenerative environment characterized by increased F4/80^+^CD206^+^ M2 macrophages, elevated IL-4 concentration, and generation of CD4^+^IL4^+^ Th2 cells within the graft [101]. These findings emphasize macrophages’ critical role in the early post-operative period in VCA, consistent with observations in solid organ transplantation [102].

### 4.4. Macrophages in Chronic Rejection of VCA

CR is a progressive form of graft injury characterized by vasculopathy, muscle and dermal fibrosis, myointimal proliferation, skin sclerosis, skin and muscle atrophy, leading to vessel narrowing and eventual ischemic damage [10]. CR is reported to affect approximately 6% of VCA recipients, with nearly 46% of cases resulting in graft loss [103]. It has become a leading cause of late VCA graft loss [104].

CR-like symptoms can be induced in preclinical rat models using major and minor MHC-mismatched donor–recipient combinations subjected to repeated acute rejection episodes and treatment cycles. In both models, increased macrophage infiltration and elevated collagen expression were observed, accompanied by enhanced intimal proliferation and vascular occlusion [12,105]. It has been hypothesized that persistent cytokine elevation following multiple rejection episodes drives macrophage infiltration and activation, which subsequently promotes fibroblast activation and collagen synthesis.

Lee et al. performed correlative multiplex immunolabeling and digital spatial proteomic profiling on a human face VCA graft that underwent CR and was removed 88 months after transplantation. The analyses revealed accelerated graft arteriosclerosis characterized by endothelialitis mediated by donor-derived CD8^+^ T cells and macrophages [106]. Notably, increased protein expression of the macrophage marker CD14 coincided with elevated CD163 levels, suggesting a role for M2 macrophages in CR of VCA. This finding aligns with studies of heart and kidney transplantation, where macrophage-derived TGF-β has been implicated in driving graft fibrosis [107,108].

### 4.5. Macrophages to Promote VCA Graft Survival

As discussed earlier, the use of pro-regenerative ECM to extend VCA survival has been associated with enhanced M2 macrophage polarization [101]. The beneficial role of M2 macrophages in promoting allograft survival has also been demonstrated in other transplantation models, including heart and islet transplantation [109,110]. Evidence suggests that M2 macrophages contribute to transplantation tolerance, potentially by modulating other regulatory cell types, such as Tr1 or Tregs [111,112].

Mregs have been explored as cellular therapeutics, as previously described. Hutchinson et al. reported the generation of TAICs from donor spleen-derived mononuclear cells using M-CSF and IFN-γ. These cells were characterized to express CD13, CD33, CD205, CD30, CD14, and CD64, confirming their macrophage identity, along with CD206 and a low level of CD80/86 [45]. The authors later classified these cells as a subset of Mregs [113]. The adoptive transfer of in vitro induced TAICs has been shown to significantly prolong graft survival in preclinical heart and lung transplantation models [49,113] and to increase the circulating Tregs in living-donor kidney transplantation recipients [114].

TAICs have also been tested in a rat hindlimb VCA model. Without any adjunctive immunosuppressive treatment, both locally and systemically administered donor-derived TAICs modestly extended graft survival by two days with statistical significance. In contrast, third-party-derived TAICs had no effect, suggesting that TAICs exert a donor-specific immunosuppressive effect [85].

## 5. Discussion and Future Directions

Although solid organ transplantation and, more recently, VCA, have significantly benefited patients suffering from end-stage organ failure and extensive tissue defects since the first successful kidney transplantation performed by Dr. Joseph Murray in 1954 [115], there remains considerable room for improvement in the long-term graft outcomes in spite of numerous advances [116,117,118]. To date, most alloimmunity research has focused on the adaptive immune system, particularly T cells involved in acute rejection. Consequently, standard immunosuppressive regimens predominantly target adaptive immunity and often insufficiently suppress components of the innate immune system, such as monocytes and macrophages, which can adversely affect graft survival [119]. In fact, widely used agents like tacrolimus and mycophenolic acid (the active metabolite of mycophenolate mofetil) exhibit significantly lower inhibitory effects on monocytes/macrophages compared to CD3^+^ T cells [120,121,122].

Given the diverse roles of the monocyte/macrophage lineage in alloimmune responses, including antigen processing and presentation, proinflammatory cytokine production and stimulation, and tissue repair, in addition to accumulating evidence implicating macrophages in IRI, acute and chronic rejection, and overall graft survival, targeted modulation of macrophage function warrants further exploration in the development of next-generation immunosuppressive strategies. For example, Dangi et al. identified the *Axl* gene as a critical regulator of monocyte differentiation to macrophages in the context of graft rejection using scRNAseq and demonstrated that targeting the *Axl* gene significantly delayed acute rejection and extended kidney graft survival [123].

According to the International Registry on Hand and Composite Tissue Transplantation (IRHCTT), 10-year survival rates for upper extremity and face VCA are 82% and 78%, respectively. Furthermore, the development of chronic rejection remains a major limitation to long-term VCA success [124]. These findings underscore the need for novel strategies that address the unique immunological challenges of VCA to further improve the graft survival rate and long-term graft quality. The above review highlights the current knowledge gaps regarding macrophage biology in VCA. The limited studies on clinical VCA patients have consistently demonstrated the correlation of elevated macrophages in acute and chronic rejection [92,93,94,106]. In addition to further delineating detailed mechanisms that macrophages employ to affect the VCA fates, targeting macrophages at various stages of VCA may offer new therapeutic opportunities to improve the transplantation outcome. Furthermore, recent advances in cell-based therapies present renewed opportunities with Mreg (TAIC)-based therapy [85]. Particularly, combining Mreg infusion with lymphodepleting or costimulatory blockade regimens may enhance tolerogenic efficacy while reducing long-term reliance on systemic immunosuppression.

As mentioned earlier, vascularized bone marrow is featured in a certain type of VCA and is equipped with tolerogenic potential [83,84], remaining healthy in tolerant VCA grafts [125]. This in-graft vascularized bone marrow is capable of hematopoiesis and can produce donor-derived monocytes and macrophages, although the presence and function of these donor-derived macrophages have not been thoroughly studied. Malone et al. demonstrated with human kidney transplant biopsies that donor-derived macrophages expressed genes involved in antigen presentation and complement signaling, whereas the recipient-derived macrophages exhibited a more proinflammatory gene profile, suggesting a division of labor between these populations [126]. In kidney transplantation, the donor-derived macrophages may be TRMs, although in the context of VCA, the interactions between donor- and recipient-derived macrophages could be complex, warranting further investigation (Figure 2). In vivo tracking of specific macrophage populations is essential for delineating the specific roles of donor-versus-recipient-originated macrophages. The superparamagnetic iron oxide nanoparticles (SPIOs) have been widely used to track macrophages with magnetic resonance imaging, offering excellent spatial resolution [127]. Furthermore, the use of subtype-specific tracers, such as CDg18 for M2 macrophages [128] or antibody-conjugated SPIOs [129], can provide deeper insights into the dynamics, spatiotemporal distribution, and functional significance of specific subtypes or cellular origin in VCA.

In recent years, extensive research in VCA has focused on graft preservation, with efforts aimed at extending graft viability from graft procurement to transplantation and minimizing IRI resulting from these processes. Different approaches, including static storage with optimized preservation solution, machine perfusion, and supercooling, have been actively explored [88,130,131]. Given the link between IRI and graft rejection [72,98,132], mitigating IRI has been shown to improve graft survival [133,134]. Modulating macrophage differentiation to attenuate IRI represents a promising strategy to improve the VCA outcomes. Furthermore, since macrophage activation precedes T cell activation during the development of rejection [98], it is reasonable to target macrophage activation at the peri-transplantation stage to prevent these cells from triggering T cell responses. One promising strategy is to supplement preservation solution with agents that suppress macrophage activation [135], such as the inhibitors of MyD88 or IRAKs, the key components in the TLR signaling pathways that activate macrophages. Exosomes derived from MSCs can also be considered, as MSCs possess immunomodulatory potential on macrophages [136,137].

There is also potential to manipulate macrophages’ function post-transplantation to improve VCA outcome. Given that current immunosuppressants are not as effective on macrophages as on T cells, as discussed previously [120,122], early supplementation with macrophage-specific activation blockers may assist in mitigating IRI and reduce the likelihood of rejection. In this context, the CSF-1R inhibitors can be considered, as CSF-1 is essential for macrophage survival and differentiation. Additionally, exploring the promotion of M2 or Mreg differentiation in macrophages following VCA could be beneficial. Similar approaches have been adopted in other transplantation models, for example, G-CSF/rapamycin-induced M2 polarization for tolerance induction in a murine islet transplantation [111]; and the generation of a specific CD11b^+^CD115^+^Gr1^+^ population with T cell-suppressive function by anti-CD40L mAb (MR1) and donor splenocyte transfusion in murine cardiac transplantation [138], supporting the potential of macrophage modulation to improve VCA outcomes.

## 6. Conclusions

Macrophages exhibit extraordinary phenotypic heterogeneity and functional versatility. In the context of VCA, they play diverse roles, spanning from ischemia–reperfusion injury to involvement in acute and chronic rejection, as well as facilitating tolerance induction. Although current research on macrophages in VCA remains limited, growing evidence highlights the substantial influence of macrophages on graft outcomes. The presence of vascularized bone marrow within certain VCA allografts adds another layer of complexity, enabling the potential contribution of donor-derived macrophage populations. Further elucidation of the dynamic roles and interactions of macrophage populations following VCA, especially during the development of rejection or tolerance, may yield novel therapeutic strategies to enhance graft survival and long-term VCA outcomes.

## Figures and Tables

**Figure 1 biomedicines-13-01425-f001:**
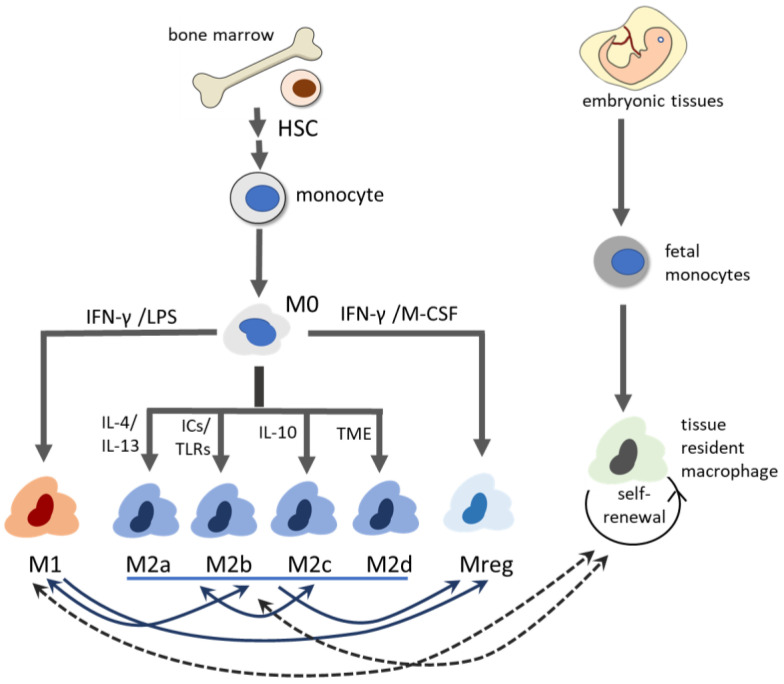
Origins and differentiation pathways of macrophages in the immune system. Macrophages originate from two primary sources: embryonic progenitors and bone marrow-derived HSCs. During embryogenesis, progenitors give rise to tissue-resident macrophages that persist into adulthood through self-renewal. In the adult, HSCs in the bone marrow differentiate into monocytes via intermediate progenitors, and subsequently into macrophages. These monocyte-derived macrophages infiltrate tissues and can adopt various phenotypes in response to local environmental cues. Solid arrows depict the dynamic phenotypic plasticity among macrophage subsets, including interconversion between M1 and M2, or among M2 subtypes. Dotted arrows represent that bone marrow-derived M1/M2 macrophages and TRMs can adopt each other’s phenotypic characteristics under certain conditions. ICs: immune complexes; TME: tumor microenvironment.

**Figure 2 biomedicines-13-01425-f002:**
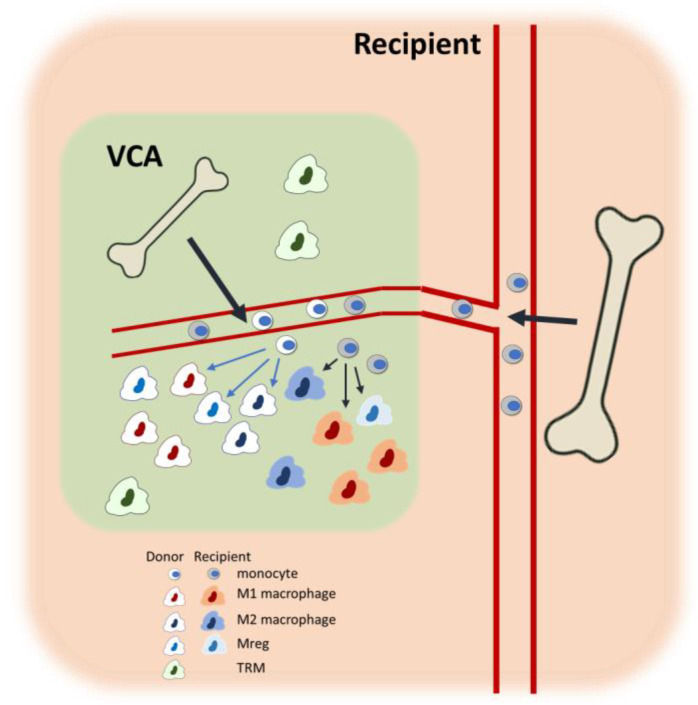
Macrophage heterogeneity in the bone marrow-containing VCA recipients. Within the VCA tissue, both donor- and recipient-derived bone marrow monocytes infiltrate and differentiate into diverse macrophage subtypes, including proinflammatory M1, anti-inflammatory M2, and Mregs. In addition, donor-derived TRMs also persist within the graft. The figure illustrates the dynamic cellular composition and highlights the contribution of both donor- and recipient-derived macrophages within the graft environment. Potential interactions and phenotypic conversions among macrophage subtypes are detailed in Figure 1.

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
