# Peer review of "Unraveling the Roles of Macrophages in Vascularized Composite Allotransplantation"

_biomedicines, 2025, doi:10.3390/biomedicines13061425_

Round 1
Reviewer 1 Report
Comments and Suggestions for Authors
Dear Editor,
Thank you for giving me the opportunity to review this article, I would like to make a few suggestions.
1-VCA, CR and macrophages are not associated with each other in the introduction section. In addition, graft vasculopathy, intimal hyperplasia and fibrosis used in defining chronic rejection should be explained in detail in the introduction section. Then, they should be associated with CR. At the end of the introduction section, write the purpose statement after a few sentences explaining the reason why VCA, CR and macrophages are discussed on the same platform and their two-way relationship with each other.
2-Please mention acute and chronic immune rejection in the abstract section, the order should be similar to the main text.
3-You should open a heading in the main text and under this heading, from general to specific, you should explain leukocytes, their subgroups and under these groups, monocytes and macrophages that differentiate from monocytes.
4-Another major concern at this point is the lack of a subheading regarding the mechanism of action of allotransplantation in the body. It is briefly mentioned in between. However, VCA needs a one or two paragraph subheading where it is explained on its own.
Reviewer 2 Report
Comments and Suggestions for Authors
It is my pleasure to extend my congratulations to the authors. The review was executed in a satisfactory manner, albeit not to a considerable extent. The subject has attracted increased attention in recent times and is considered to be of interest. This phenomenon is likely to be of concern to clinicians.
It is recommended that the authors introduce further information on clinical cases in the field in the discussion section.
Round 2
Reviewer 1 Report
Comments and Suggestions for Authors
Dear Editor and Authors,
I am witnessing a magnificent review article describing the roles of macrophages in vascularized Compo-2-site allotransplantation. I have previously reviewed this article and made many revisions. I see that the authors have made these revisions carefully and improved their work. The detailed description of macrophages has provided a significant gain to the literature, I appreciate the authors in this respect, I thank the editor for giving me this special task, best regards